# Prevalence and associated factors of overweight and obesity among Afghan school children: A cross-sectional analytical study from Kandahar City, Afghanistan

**Bilal Ahmad Rahimi** [1]*, **Aziz Ahmad Khalid**[2], **Wahid Ahmad Khalid**[3],
**Javed Ahmad Rahimi**[4], **Walter R. Taylor**[5]

1 Department of Pediatrics, Faculty of Medicine, Kandahar University, Kandahar, Afghanistan,
2 Department of Economics, Jamia Millia Islamia, Central University, New Delhi, India, 3 Department of Economics, Savitribai Phule Pune University, Pune, Maharashtra, India, 4 Department of Business Administration, Gujarat University, Ahmedabad, Gujarat, India, 5 Mahidol Oxford Tropical Medicine Clinical Research Unit (MORU), Mahidol University, Bangkok, Thailand

* drbilal77@yahoo.com

## Abstract

### Background

Childhood overweight and obesity is an emerging public health problem in developing countries. This is the first school-based study of its type from Afghanistan to estimate the prevalence and associated factors of overweight and obesity among Afghan school children aged 6–18 years in Kandahar City of Afghanistan.

### Methods

This cross-sectional analytical study was conducted among 2281 school children from January 10–April 15, 2024. Sociodemographic properties, anthropometric measurements, and other data were collected from all the participants. Data were analyzed using descriptive statistics, Chi-square test, and multiple logistic regression analysis.

### Results

Among the 2281 enrolled children, 72.5% were boys, 65.1% going to private schools, and 53.8% poor. The prevalence of overweight and obesity was 11.5% (6.6% were overweight and 4.9% were obese). The mean (SD) age was 12.7 (2.1) years. By logistic regression analysis, statistically significant risk factors associated with overweight and obesity were being boy (AOR 1.5 and 95% CI 1.1–2.0), student of private school (AOR 2.2 and 95% CI 1.5–2.8), belonging to a rich family (AOR 1.9 and 95% CI 1.3–2.7), and parental obesity (AOR 1.5 and 95% CI 1.1–2.0).

### Conclusion

School children of Kandahar city are suffering from overweight/obesity. Overweight/obesity should be dealt with as an emerging problem in school children of Kandahar city. It

**Data availability statement:** All relevant data are within the manuscript.

**Funding:** This study did not receive any specific funding. W. R. Taylor is part-funded by Wellcome under grant 220211. For the purposes of Open Access, the author has applied a CC BY public copyright license to any Author Accepted Manuscript version arising from this submission.

**Competing interests:** All the authors do not have any competing interests.

**Abbreviations:** AOR, Adjusted odds ratio; BAZ, BMI-for-age Z score; BMI, Body mass index; CI, Confidence interval; COR, Crude odds ratio; HAZ, height-for-age Z score; SD, Standard deviation; SPSS, Statistical Package for the Social Sciences; UNICEF, United Nations Children's Fund; USD, United States Dollar; WFP, World Food Program; WHO, World Health Organization; WAZ, Weight-for-age Z score; WHZ, Weight-for-height Z score.

is recommended that Afghanistan ministries of education and public health, with the help of international donor agencies, such as WHO and UNICEF, work together in controlling overweight and obesity in school children of Kandahar city. Periodic special awareness programs on the prevention and control of overweight/obesity should be conducted in schools, radio, television, and other sources of social media.

## Introduction

Globally, there has been a significant increase in the prevalence of overweight and obesity among primary school children in both developed and developing countries [1–3]. In 2016, worldwide 124 million children aged 5–19 years were estimated to be suffering from obesity [4]. A study conducted in the USA revealed that 31.0% of the children were overweight [5]. A study conducted in Italy showed that the prevalence of overweight/obesity among children was 25.2% [6]. In India, the prevalence of overweight among boys and girls was 18% and 16%, respectively [7].

Childhood overweight and obesity is an emerging public health problem in developing countries [8]. This is largely related to the increasing westernization of societies and associated changes in lifestyle [9]. Many developing countries are currently in a state of nutritional transition, i.e., there is a persistently increased prevalence of under-nutrition in the face of emerging overweight and obesity [10,11]. Overweight/obesity causes complications during childhood and adolescence which persist into adulthood, increasing the risk of morbidity and mortality later in life [12]. These complications are the development of high blood pressure, long-term cardiovascular morbidities, and early death [8].

Some studies have reported that environmental and genetic factors affect the prevalence of overweight and obesity, although some inconsistencies in the effects have been reported [13,14]. Major risk factors for overweight/obesity among children, especially in developing countries, are belonging to rich families, higher levels of maternal education, decreased physical activity, female gender, and race [15,16]. In childhood, overweight and obesity are amenable to treatment. So, early intervention could help in reducing both short- and long-term complications of overweight and obesity, as well as break the tracking into adolescence and adulthood [17].

Very few studies among children have been reported from the Kandahar province of Afghanistan [18–23]. To our knowledge, regarding overweight and obesity among adolescents, only two studies have been reported from the entire Afghanistan. In 2014, the Global School-based Student Health Survey (GSHS) reported that the prevalence of overweight was 19.0% among 1493 students aged 12–15 years [24]. However, the Afghanistan National Nutrition Survey 2013 revealed that 10.4% of 6392 adolescent girls aged 10–19 years were overweight or obese in Afghanistan [25].

To the level of our knowledge, there is no published study from the entire of Afghanistan that studies the prevalence and associated factors of overweight and obesity among school children aged 6–18 years. So, this is the first school-based study of its type not only from Kandahar city but the whole of Afghanistan to estimate the prevalence and associated factors of overweight and obesity among Afghan school children in Kandahar city of Afghanistan.

## Materials and methods

### Study design and study area

This was a school-based cross-sectional analytical study conducted between January 10 and April 15, 2024 in Kandahar City. This city of southwest Afghanistan has an altitude of 1010

meters above sea level and a population of approximately 614118 people. Among the 145 schools in Kandahar City, 10 schools (five government and five private schools) were randomly selected for this study using the lottery method of randomization.

## Study population and sample size calculation

The study population of this research was composed of school children aged 6–18 years and permanent residents of Kandahar. All those children were excluded from the study whose parent/guardian refused to participate in the study. The demographic characteristics of the excluded children were similar to the included children. Also, during the selection process, socioeconomic status, diet, distance between home to school, and other factors that may affect the overweight/obesity rate were taken into consideration. So, there was no representative bias in this study.

For this study, the sample size was calculated using the Epi Info software version 7.2.2.6 (CDC, Atlanta, Georgia, USA). The two-sided confidence level was taken as 95%, while the power of the study was 90%. A non-response rate of 20% was added. So, our sample size was 2395 school children, with 240 study participants selected from each school. Among these children, 114 (4.8%) were excluded because their parents/guardians refused to take part in the study. The response rate was 95.2%. So, we collected data from 2281 school children.

## Sample collection, data quality management, and anthropometric measurements in children

For this study, children from the 10 schools were selected using the lottery method of randomization. A researcher-made questionnaire was used in two local languages (Pashto and Dari) with questions regarding socio-demographic characteristics, anthropometric measurements, and other characteristics for the study. Paper-based data collection was conducted by experienced and trained researchers. In each school, before starting the data collection, all children and their parents/guardians were provided with short information about the objectives and methods of this study. All the data collectors were well trained on how to conduct the interview with school children and their parents/guardians and how to do the anthropometric measurements. Data of the young age school children who were not able to provide complete information in the interview were collected from their parents/guardians. In addition, we also involved their parents/guardians in the interviews regarding the variables that children could not answer well, such as family economic status, information about parents (obesity, hypertension, diabetes mellitus, etc.), and eating habits of the children.

All methods were performed following the relevant guidelines and regulations. Data of anthropometric measurements such as body weight and height were collected from each child. For height, all study participants stood barefoot against a vertical wall, and height was measured using a stadiometer to the nearest 0.1 cm. The body weight of the study participants with lightweight clothes was measured to the nearest 0.1 kg with a validated digital balance. For quality control, approximately 10% of the study participants were randomly selected and measured by an experienced researcher who was blinded by the results of previous measurements. Besides, data was double entered to minimize the errors.

In this study, poverty was defined as a family that earns < 155 Afghanis (<2.15 USD) per person per day, based on the definition by the World Bank [26]. For nutritional status, anthropometric indices such as height-for-age Z (HAZ) score, weight-for-age Z (WAZ) score, weight-for-height Z (WHZ) score, and BMI-for-age Z (BAZ) score were computed as per the 2009 WHO Child Growth Standards median to assess the growth and nutritional status of the children [27]. Children were defined as normal if their WAZ score was −2 SD to +2 SD or

HAZ score −2 SD to +2 SD or BAZ score −2 SD to +1 SD of WHO Child Growth Standards median. Children were defined as undernourished if their WAZ score was < −2 SD (underweight) or HAZ score < −2 SD (stunting) or BAZ score < −2 SD (thinness) of the WHO Child Growth Standards median. Children were defined as overweight if their BAZ score was> +1 SD of the WHO Child Growth Standards median. Children were defined as obese if their BAZ score was> +2 SD of the WHO Child Growth Standards median [27].

### Ethical considerations

Ethical approval was taken from the Ethics Committee of Kandahar University (code number KDRU-EC-2023.17) and permission from the officials of the Kandahar Province Education Department. Before the data collection, written informed consents and assents were obtained from the parents/guardians and children, respectively. Interviews of the study participants using a predesigned questionnaire were conducted. For data collection, only children's initials were used. Information on the participants would not be disclosed. Before entering into the computer for analysis, the collected data were coded and de-identified.

### Data analysis

The data were entered into Microsoft Excel 2021, cleaned, and imported to Statistical Package for the Social Sciences (SPSS) version 25 (Chicago, IL, USA) for statistical analysis. Independent variables in this study included sociodemographic factors, physical activity, and food consumption patterns of school children. Descriptive analysis including frequency, percentage, mean, standard deviation (SD), and range was used to summarize independent variables and the prevalence of overweight/obesity among school children. A Chi-square test (using crude odds ratio [COR]) was performed to find the association between overweight/obesity and independent variables (suspected risk factors). All the independent variables that were statistically significant in Chi-square analysis were adjusted and assessed for independence in multiple logistic regression (using adjusted odds ratio [AOR]) to determine the risk factors of overweight/obesity among school children. A $p$-value of < 0.05 was considered statistically significant.

## Results

This study was conducted during January–April 2024 in 2281 Afghan school children with a mean (SD) age of 12.7 (2.1) years. Of these children, 72.5% were boys, 65.1% were in private schools, and 53.8% had a family monthly income of < 10,000 Afghanis. Parental obesity was present in 28.6% of children while 67.2% of children went to and from school using a motorcycle, car, or bus. Among these children, the prevalence of obesity, overweight, underweight, and normal BMI were present in 4.9%, 6.6%, 51.7%, and 36.8%, respectively (Table 1).

A comparison of main sociodemographic and other characteristics of the school children between government vs private schools are described in Table 2. Statistically significant differences between government and private school students were observed as follows: most of the private school students were adolescents, were girls, living in rented houses, belonged to rich families, had parental obesity, used motorcycle/car/bus to go to and from school, spent ≥ 2 hours daily on watching television, sometimes/rarely consumed fruits, and were obese. Based on Chi-square test analysis, statistically significant factors associated with overweight/obesity among school children were (i) being a boy (COR 1.4, 95% CI 1.0–1.9, and $p$-value 0.027), (ii) student of private school (COR 2.1, 95% CI 1.5–2.8, and $p$-value < 0.001), (iii) belonging to rich family (COR 2.1, 95% CI 1.5–3.0, and $p$-value < 0.001), (iv) parental obesity (COR 1.4, 95% CI 1.1–1.9, and $p$-value 0.013), (v) using motorcycle/car/bus for transport to

**Table 1. Sociodemographic, physical activities, and food consumption patterns of the study participants with the comparison between boys and girls.**

| Variable | Frequency (%), n = 2281 | Boys n = 1654 | Girls n = 627 | P-value |
|---|---|---|---|---|
| **Age** | | | | |
| 6–11 years | 882 (38.7) | 636 (72.1) | 246 (27.9) | 0.732 |
| 12–18 years | 1399 (61.3) | 1018 (72.8) | 381 (27.2) | |
| **School type** | | | | |
| Government | 797 (34.9) | 636 (79.8) | 161 (20.2) | <0.001 |
| Private | 1484 (65.1) | 1018 (68.6) | 466 (31.4) | |
| **Residence ownership** | | | | |
| Owned | 1542 (67.6) | 1061 (68.8) | 481 (31.2) | <0.001 |
| Rented | 739 (32.4) | 593 (80.2) | 146 (19.8) | |
| **Family economic status** | | | | |
| Not rich | 2065 (90.5) | 1489 (72.1) | 576 (27.9) | 0.180 |
| Rich | 216 (9.5) | 165 (76.4) | 51 (23.6) | |
| **Father's literacy** | | | | |
| Literate | 471 (20.6) | 368 (78.1) | 103 (21.9) | 0.002 |
| Illiterate | 1810 (79.4) | 1286 (71.0) | 524 (29.0) | |
| **Mother's literacy** | | | | |
| Literate | 420 (18.4) | 274 (65.2) | 146 (34.8) | <0.001 |
| Illiterate | 1861 (81.6) | 1380 (74.2) | 481 (25.8) | |
| **Father occupation** | | | | |
| Employed | 1989 (87.2) | 1459 (73.4) | 530 (26.6) | 0.019 |
| Unemployed | 292 (12.8) | 195 (66.8) | 97 (33.2) | |
| **Mother occupation** | | | | |
| Employed | 78 (3.4) | 58 (74.4) | 20 (25.6) | 0.710 |
| Unemployed/Housewife | 2203 (96.6) | 1596 (72.4) | 607 (27.6) | |
| **Parental obesity** | | | | |
| Yes | 652 (28.6) | 505 (77.5) | 147 (22.5) | 0.001 |
| No | 1629 (71.4) | 1149 (70.5) | 480 (29.5) | |
| **Parental hypertension** | | | | |
| Yes | 198 (8.7) | 162 (81.8) | 36 (18.2) | 0.002 |
| No | 2083 (91.3) | 1492 (71.6) | 591 (28.4) | |
| **Parental diabetes mellitus** | | | | |
| Yes | 70 (3.1) | 48 (68.6) | 22 (31.4) | 0.453 |
| No | 2211 (96.9) | 1606 (72.6) | 605 (27.4) | |
| **Modes of transport to and from school** | | | | |
| Walking/Bicycle | 542 (23.8) | 386 (71.2) | 156 (28.8) | 0.440 |
| Motorcycle/car/bus | 1739 (76.2) | 1268 (72.9) | 471 (27.1) | |
| **Doing exercise/playing outdoor game** | | | | |
| Always/mostly | 724 (31.7) | 521 (72.0) | 203 (28.0) | 0.688 |
| Sometimes/rarely | 1557 (68.3) | 1133 (72.8) | 424 (27.2) | |
| **Time spent daily on watching television** | | | | |
| < 2 hours | 419 (18.4) | 302 (72.1) | 117 (27.9) | 0.825 |
| ≥ 2 hours | 1862 (81.6) | 1352 (72.6) | 510 (27.4) | |
| **Consumption of snacks in between meals** | | | | |
| Always/mostly | 1318 (57.8) | 949 (72.0) | 369 (28.0) | 0.524 |
| Sometimes/rarely | 963 (42.2) | 705 (73.2) | 258 (26.8) | |

*(Continued)*

**Table 1.** (Continued)

| Variable | Frequency (%), n = 2281 | Boys n = 1654 | Girls n = 627 | P-value |
|---|---|---|---|---|
| **Eating confectionery/sweet foods** | | | | |
| Always/mostly | 575 (25.2) | 412 (71.7) | 163 (28.3) | 0.593 |
| Sometimes/rarely | 1706 (74.8) | 1242 (72.8) | 464 (27.2) | |
| **Drinking soft drinks** | | | | |
| Always/mostly | 1481 (64.9) | 1085 (73.3) | 396 (26.7) | 0.275 |
| Sometimes/rarely | 800 (35.1) | 569 (71.1) | 231 (28.9) | |
| **Frequency of consuming vegetables** | | | | |
| Always/mostly | 1273 (55.8) | 921 (72.3) | 352 (27.7) | 0.844 |
| Sometimes/rarely | 1008 (44.2) | 733 (72.7) | 275 (27.3) | |
| **Frequency of consuming fruit** | | | | |
| Always/mostly | 362 (15.9) | 292 (80.7) | 70 (19.3) | <0.001 |
| Sometimes/rarely | 1919 (84.1) | 1362 (71.0) | 557 (29.0) | |
| **Overweight/obesity** | | | | |
| Yes | 262 (11.5) | 205 (78.2) | 57 (21.8) | 0.027 |
| No | 2019 (88.5) | 1449 (71.8) | 570 (28.2) | |

and from school (COR 1.8, 95% CI 1.3–2.6, and *p*-value 0.001), and (vi) spending ≥ 2 hours daily on watching television (COR 1.7, 95% CI 1.2–2.5, and *p*-value 0.006). Finally, based on logistic regression analysis, the independent statistically significant risk factors associated with overweight/obesity among school children were (i) being boy (AOR 1.5, 95% CI 1.1–2.0, and *p*-value 0.021), (ii) student of private school (AOR 2.2, 95% CI 1.5–2.8, and *p*-value < 0.001), (iii) belonging to rich family (AOR 1.9, 95% CI 1.3–2.7, and *p*-value 0.001), (iv) parental obesity (AOR 1.5, 95% CI 1.1–2.0, and *p*-value 0.006) (Table 3).

## Discussion

In this study, the prevalence of overweight and obesity among 2281 recruited school children of Kandahar city was 11.5%. Compared to this study, increased prevalence of overweight and obesity was present in Brazil (prevalence was 32.4% among 1125 school children aged 5.6–18 years) [3], China (prevalence was 28.2% among 3140 school children aged 7–18 years) [28], and Nepal (prevalence was 25.7% among 575 school children aged 6–13 years) [2]. Similarly, increased prevalence was reported from Tunisia (prevalence was 23.3% among 2872 boys aged 15–19 years) [29], Tanzania (prevalence was 22.6% among 1718 urban primary school children aged 8–13 years) [30], and Uganda (prevalence was 20% among 422 urban primary school children aged 7–17 years) [31]. However, a study conducted in Nigeria reported that the prevalence of overweight and obesity was 4.9% among 1187 semi-urban primary school children aged 6–11 years [8]. The inconsistence in the prevalence of overweight and obesity among these studies may be explained by differences in the study populations. It is stated that the demography of overweight/obesity prevalence differs with culture, structure, and ecology of the environment of the study populations [30].

In this study, being boy was a statistically significant risk factor for overweight/obesity in school children. Similar results have been reported in studies conducted in Iran (among 1000 urban school children aged 7–12 years) [32], China (among 3140 school children aged 7–18 years) [28], Nepal (among 575 school children aged 6–13 years) [2], and Australia (among 5252 school students aged 12–18 years) [33]. However, being girl has been reported as risk

**Table 2. Main sociodemographic and other characteristics of the study participants with the comparison between government and private school students.**

| Variable | Frequency n (%) n = 2281 | Government school n (%) n = 797 | Private school n (%) n = 1484 | *P*-value |
|---|---|---|---|---|
| **Age** | | | | |
| 6–11 years | 882 (38.7) | 332 (37.6) | 550 (62.4) | 0.032 |
| 12–18 years | 1399 (61.3) | 465 (33.2) | 934 (66.8) | |
| **Sex** | | | | |
| Boys | 1654 (72.5) | 636 (38.5) | 1018 (61.5) | <0.001 |
| Girls | 627 (27.5) | 161 (25.7) | 466 (74.3) | |
| **Residence ownership** | | | | |
| Owned | 1542 (67.6) | 574 (37.2) | 968 (62.8) | 0.001 |
| Rented | 739 (32.4) | 223 (30.2) | 516 (69.8) | |
| **Family economic status** | | | | |
| Not rich | 2065 (90.5) | 739 (35.8) | 1326 (64.2) | 0.009 |
| Rich | 216 (9.5) | 58 (26.9) | 158 (73.1) | |
| **Father's literacy** | | | | |
| Literate | 471 (20.6) | 158 (33.5) | 313 (66.5) | 0.476 |
| Illiterate | 1810 (79.4) | 639 (35.3) | 1171 (64.7) | |
| **Mother's literacy** | | | | |
| Literate | 420 (18.4) | 163 (38.8) | 257 (61.2) | 0.066 |
| Illiterate | 1861 (81.6) | 634 (34.1) | 1227 (65.9) | |
| **Father occupation** | | | | |
| Employed | 1989 (87.2) | 690 (34.7) | 1299 (65.3) | 0.513 |
| Unemployed | 292 (12.8) | 107 (36.6) | 185 (63.4) | |
| **Mother occupation** | | | | |
| Employed | 78 (3.4) | 34 (43.6) | 44 (56.4) | 0.103 |
| Unemployed/Housewife | 2203 (96.6) | 763 (34.6) | 1440 (65.4) | |
| **Parental obesity** | | | | |
| Yes | 652 (28.6) | 266 (40.8) | 386 (59.2) | <0.001 |
| No | 1629 (71.4) | 531 (32.6) | 1098 (67.4) | |
| **Parental hypertension** | | | | |
| Yes | 198 (8.7) | 68 (34.3) | 130 (65.7) | 0.854 |
| No | 2083 (91.3) | 729 (35.0) | 1354 (65.0) | |
| **Parental diabetes mellitus** | | | | |
| Yes | 70 (3.1) | 26 (37.1) | 44 (62.9) | 0.695 |
| No | 2211 (96.9) | 771 (34.9) | 1440 (65.1) | |
| **Modes of transport to and from school** | | | | |
| Walking/Bicycle | 542 (23.8) | 259 (47.8) | 283 (52.2) | <0.001 |
| Motorcycle/car/bus | 1739 (76.2) | 538 (30.9) | 1201 (69.1) | |
| **Doing exercise/playing outdoor game** | | | | |
| Always/mostly | 724 (31.7) | 257 (35.5) | 467 (64.5) | 0.704 |
| Sometimes/rarely | 1557 (68.3) | 540 (34.7) | 1017 (65.3) | |
| **Time spent daily on watching television** | | | | |
| < 2 hours | 419 (18.4) | 212 (50.6) | 207 (49.4) | <0.001 |
| ≥ 2 hours | 1862 (81.6) | 585 (31.4) | 1277 (68.6) | |
| **Consumption of snacks in between meals** | | | | |
| Always/mostly | 1318 (57.8) | 453 (34.4) | 865 (65.6) | 0.504 |
| Sometimes/rarely | 963 (42.2) | 344 (35.7) | 619 (64.3) | |

*(Continued)*

**Table 2.** (Continued)

| Variable | Frequency | Government school n (%) | Private school | P-value |
|---|---|---|---|---|
| | n (%) | n = 797 | n (%) | |
| | n = 2281 | | n = 1484 | |
| **Eating confectionery/sweet foods** | | | | |
| Always/mostly | 575 (25.2) | 209 (36.3) | 366 (63.7) | 0.413 |
| Sometimes/rarely | 1706 (74.8) | 588 (34.5) | 1118 (65.5) | |
| **Drinking soft drinks** | | | | |
| Always/mostly | 1481 (64.9) | 520 (35.1) | 961 (64.9) | 0.816 |
| Sometimes/rarely | 800 (35.1) | 277 (34.6) | 523 (65.4) | |
| **Frequency of consuming vegetables** | | | | |
| Always/mostly | 1273 (55.8) | 457 (35.9) | 816 (64.1) | 0.281 |
| Sometimes/rarely | 1008 (44.2) | 340 (33.7) | 668 (66.3) | |
| **Frequency of consuming fruit** | | | | |
| Always/mostly | 362 (15.9) | 149 (41.2) | 213 (58.8) | 0.007 |
| Sometimes/rarely | 1919 (84.1) | 648 (33.8) | 1271 (66.2) | |
| **Overweight/obesity** | | | | |
| Yes | 262 (11.5) | 57 (21.8) | 205 (78.2) | <0.001 |
| No | 2019 (88.5) | 740 (36.7) | 1279 (63.3) | |

factor of overweight/obesity in studies from Tanzania (among 1718 urban primary school children aged 8–13 years) [30], Nigeria (among 1187 semi-urban primary school children aged 6–11 years) [8], Ghana (among 270 primary school children aged 5–15 years) [34], and Congo (among 486 adolescents) [35]. In many societies, this could be due to the restrictions on the physical activity of girls by parents [36,37]. Also, puberty causes sexual dimorphism due to which boys get more muscle-bone mass while girls gain more fat mass [38]. Among girls, early puberty is usually associated with shorter height, higher BMI, and a higher risk of obesity in adulthood [38].

In this study, being a student from a rich family was a risk factor for overweight/obesity. This is in accordance with the results of studies conducted in Nigeria (among 1187 semi-urban primary school children aged 6–11 years) [8], Ghana (among 270 primary school children aged 5–15 years) [34], Tunisia (among 1295 boys aged 15–19 years) [29], and Brazil (prevalence was 32.4% among 1125 school children aged 5.6–18 years) [3]. Contrary to this, a study in the United States revealed that the prevalence of overweight decreased with increasing income among 7135 non-Hispanic white Americans aged 12–20 years [39]. Also, a study in Norway among 15966 school students aged 15–16 years showed that poor family economy was significantly associated with overweight and obesity [40]. The above-mentioned differences in the prevalence patterns could be due to differences in the lifestyle between developed and developing countries [41]. In developing countries, as compared to poor families, rich families have better access to meat and other energy-dense fast foods, which are more expensive than other foods like vegetables [41]. Fast foods are more popular due to easy accessibility, sweet and better taste, occupation of the parents, increased socio-economic status of the family, and effects of social media [42]. Contrary to this, in developed countries, upper socio-economic class families have increased awareness of the burden of obesity and its association with unhealthy diets, usually consuming more fruits and vegetables than lower socio-economic class families [41]. Also, children from rich families mostly use cars for transportation, do less or no housework, and do less active work, which in turn is associated with a higher risk of developing childhood obesity [28].

**Table 3. Chi-square and logistic regression analysis of the factors associated with overweight/obesity among the study participants.**

| Variable | Total n (%), n = 2281 | Overweight/obesity | | COR (95% CI) | *P*-value | AOR (95% CI)[a] | *P*-value |
|---|---|---|---|---|---|---|---|
| | | Present n (%), n = 262 | Absent n (%), n = 2019 | | | | |
| **Age** | | | | | | | |
| 6–11 years | 882 (38.7) | 105 (11.9) | 777 (88.1) | 0.9 (0.7–1.2) | 0.619 | – | – |
| 12–18 years | 1399 (61.3) | 157 (11.2) | 1242 (88.8) | 1 | | | |
| **Sex** | | | | | | | |
| Boys | 1654 (72.5) | 205 (12.4) | 1449 (87.6) | 1.4 (1.0–1.9) | 0.027 | 1.5 (1.1–2.0) | 0.021 |
| Girls | 627 (27.5) | 57 (9.1) | 570 (90.9) | 1 | | 1 | |
| **School type** | | | | | | | |
| Government | 797 (34.9) | 57 (7.2) | 740 (92.8) | 1 | <0.001 | 1 | <0.001 |
| Private | 1484 (65.1) | 205 (13.8) | 1279 (86.2) | 2.1 (1.5–2.8) | | 2.2 (1.5–2.8) | |
| **Residence ownership** | | | | | | | |
| Owned | 1542 (67.6) | 175 (11.3) | 1367 (88.7) | 1 | 0.766 | – | – |
| Rented | 739 (32.4) | 87 (11.8) | 652 (88.2) | 1.0 (0.8–1.4) | | | |
| **Family economic status** | | | | | | | |
| Not rich | 2065 (90.5) | 219 (10.6) | 1846 (89.4) | 1 | <0.001 | 1 | 0.001 |
| Rich | 216 (9.5) | 43 (19.9) | 173 (80.1) | 2.1 (1.5–3.0) | | 1.9 (1.3–2.7) | |
| **Father's literacy** | | | | | | | |
| Literate | 471 (20.6) | 37 (7.9) | 434 (92.1) | 1 | 0.006 | – | – |
| Illiterate | 1810 (79.4) | 225 (12.4) | 1585 (87.6) | 0.6 (0.4–0.9) | | | |
| **Mother's literacy** | | | | | | | |
| Literate | 420 (18.4) | 42 (10.0) | 378 (90.0) | 1 | 0.290 | – | – |
| Illiterate | 1861 (81.6) | 220 (11.8) | 1641 (88.2) | 1.2 (0.9–1.7) | | | |
| **Father occupation** | | | | | | | |
| Employed | 1989 (87.2) | 230 (11.6) | 1759 (88.4) | 1.1 (0.7–1.6) | 0.762 | – | – |
| Unemployed | 292 (12.8) | 32 (11.0) | 260 (89.0) | 1 | | | |
| **Mother occupation** | | | | | | | |
| Employed | 78 (3.4) | 6 (7.7) | 72 (92.3) | 1 | 0.285 | – | – |
| Unemployed/Housewife | 2203 (96.6) | 256 (11.6) | 1947 (88.4) | 1.6 (0.7–3.7) | | | |
| **Parental obesity** | | | | | | | |
| Yes | 652 (28.6) | 92 (14.1) | 560 (85.9) | 1.4 (1.1–1.9) | 0.013 | 1.5 (1.1–2.0) | 0.006 |
| No | 1629 (71.4) | 170 (10.4) | 1459 (89.6) | 1 | | 1 | |
| **Parental hypertension** | | | | | | | |
| Yes | 198 (8.7) | 30 (15.2) | 168 (84.8) | 1.4 (0.9–2.2) | 0.091 | – | – |
| No | 2083 (91.3) | 232 (11.1) | 1851 (88.9) | 1 | | | |
| **Parental diabetes mellitus** | | | | | | | |
| Yes | 70 (3.1) | 5 (7.1) | 65 (92.9) | 1 | 0.247 | – | – |
| No | 2211 (96.9) | 257 (11.6) | 1954 (88.4) | 1.7 (0.7–4.3) | | | |
| **Modes of transport to and from school** | | | | | | | |
| Walking/Bicycle | 542 (23.8) | 40 (7.4) | 502 (92.6) | 1 | 0.001 | 1 | 0.023 |
| Motorcycle/car/bus | 1739 (76.2) | 222 (12.8) | 1517 (87.2) | 1.8 (1.3–2.6) | | 1.5 (1.1–2.2) | |
| **Doing exercise/playing outdoor game** | | | | | | | |
| Always/mostly | 724 (31.7) | 97 (13.4) | 627 (86.6) | 0.8 (0.6–1.0) | 0.051 | – | – |
| Sometimes/rarely | 1557 (68.3) | 165 (10.6) | 1392 (89.4) | 1 | | | |

*(Continued)*

**Table 3.** (Continued)

| Variable | Total n (%), n = 2281 | Overweight/obesity | | COR (95% CI) | *P*-value | AOR (95% CI)[a] | *P*-value |
|---|---|---|---|---|---|---|---|
| | | Present n (%), n = 262 | Absent n (%), n = 2019 | | | | |
| **Time spent daily on watching television** | | | | | | | |
| < 2 hours | 419 (18.4) | 32 (7.6) | 387 (92.4) | 1 | 0.006 | 1 | 0.063 |
| ≥ 2 hours | 1862 (81.6) | 230 (12.4) | 1632 (87.6) | 1.7 (1.2–2.5) | | 1.5 (1.0–2.2) | |
| **Consumption of snacks in between meals** | | | | | | | |
| Always/mostly | 1318 (57.8) | 152 (11.5) | 1166 (88.5) | 1.0 (0.8–1.3) | 0.935 | – | – |
| Sometimes/rarely | 963 (42.2) | 110 (11.4) | 853 (88.6) | 1 | | | |
| **Eating confectionery/sweet foods** | | | | | | | |
| Always/mostly | 575 (25.2) | 60 (10.4) | 515 (89.6) | 1 | 0.361 | – | – |
| Sometimes/rarely | 1706 (74.8) | 202 (11.8) | 1504 (88.2) | 1.2 (0.9–1.6) | | | |
| **Drinking soft drinks** | | | | | | | |
| Always/mostly | 1481 (64.9) | 149 (10.1) | 1332 (89.9) | 0.7 (0.5–0.9) | 0.004 | – | – |
| Sometimes/rarely | 800 (35.1) | 113 (14.1) | 687 (85.9) | 0.8 (0.6–1.0) | | | |
| **Frequency of consuming vegetables** | | | | | | | |
| Always/mostly | 1273 (55.8) | 162 (12.7) | 1111 (87.3) | 1 | 0.037 | – | – |
| Sometimes/rarely | 1008 (44.2) | 100 (9.9) | 908 (90.1) | | | | |
| **Frequency of consuming fruit** | | | | | | | |
| Always/mostly | 362 (15.9) | 41 (11.3) | 321 (88.7) | 1 | 0.917 | – | – |
| Sometimes/rarely | 1919 (84.1) | 221 (11.5) | 1698 (88.5) | 1.0 (0.7–1.5) | | | |

AOR, Adjusted Odds Ratio; CI, confidence interval; COR, crude odds ratio; N, number.

a Variables with non-significance in Chi-square analysis were not included in logistic regression model.

In this study, statistically significant increased overweight/obesity was observed in students of private schools. Similar results have been reported in studies conducted in Uganda (among 422 urban primary school children aged 7–17 years) [31], Tanzania (among 1718 urban primary school children aged 8–13 years) [30], Nigeria (among 1187 semi-urban primary school children aged 6–11 years) [8], and Iran (among 1000 urban school children aged 7–12 years) [32]. Rich families usually provide more daily pocket money to their school-going children. This enables these children to eat an extra meal of junk food at school. Also, this might be related to the fact that rich parents usually send their children to private schools in most of the developing countries [43].

In this study, a statistically significant increase in overweight and obesity was observed in school children with parental obesity. A study conducted in China among 3140 school children aged 7–18 years also reported that overweight and obesity were significantly more among school students with parental obesity [28]. Genetics plays an important role in obesity. So, obesity usually tracks in families. Having obese parents increases one's risk for obesity, even if family members do not live together or share the same patterns of exercise and food intake [44–47]. However, many studies have revealed that besides the high genetic similarity among family members, parents play a crucial role in the development of children's physical activity patterns [48], eating behavior, and attitudes [49].

In this study, a statistically significant increased prevalence of overweight and obesity was present among children whose mode of transportation to and from school was bus, car, or motorcycle than those children who were walking. This is in accordance with the studies

conducted in Uganda (among 422 urban primary school children aged 7–17 years) [31], Nepal (among 575 school children aged 6–13 years) [2], Ghana (among 400 primary school children aged 6–12 years) [50], and China (among 3140 school children aged 7–18 years) [28]. Contrary to the results of this study, studies conducted in Ghana (among 400 school-age children) [51] and Portugal (among 1786 school children aged 6–13 years) [52] did not find any significant association between modes of transportation to school and overweight/obesity among primary school children. Health promotion interventions should address childhood overweight and obesity to promote a culture of walking as a good practice among primary school children. Both parents and children should be sensitized about the health benefits of walking compared to bus, car, or motorcycle. Also, roads should have pedestrian lanes, which will enable children to safely walk to and from school [31].

## Strengths and limitations

The first and main strength of this study is that it is the first study of its type not only from Kandahar city but from the entire Afghanistan to investigate the prevalence and risk factors of overweight and obesity among school children. Second, data was randomly collected from ten (five government and five private) schools helped us in generalizing the results of this study to the school students of the entire Kandahar City.

This study had several limitations. First, this study was conducted among school children of Kandahar city. So, its findings may not reflect the overall scenario of the entire country and may not be generalized to the school children in rural areas. Second, information about variables related to time was answered from memory. So, recall bias cannot be ruled out. Third, as this was a cross-sectional, a causal link between the variables could not be determined. Nevertheless, this study reported probable risk factors for overweight and obesity in school children, which can further be studied in the future by longitudinal studies.

## Conclusions

The prevalence of overweight and obesity was 11.5% among school children of Kandahar city. The main factors associated with overweight and obesity in school children were being boy, from rich family, student of private school, and having parental obesity. In Afghanistan, the currently focused issue is only undernutrition. However, based on the results of this study, overweight/obesity should be dealt with as an emerging problem in school children of Kandahar City.

It is recommended that Afghanistan ministries of education and public health, with the help of international donor agencies, such as WHO and UNICEF, work together in controlling overweight and obesity alongside the currently focused undernutrition in school children of Kandahar city. Periodic special awareness programs on the prevention and control of overweight/obesity should be conducted in schools, radio, television, and other sources of social media. Further studies are needed to be conducted in school children of rural areas in Kandahar province. Also, the conduction of nationwide multicenter longitudinal studies in all 34 provinces of Afghanistan is crucial to see the prevalence of overweight/obesity not only among school children but also in adults and elderly populations.

## Acknowledgments

We are sincerely thankful to the authorities of Kandahar University, the Kandahar Directorate of Education, and the enrolled schools. We are also thankful to all the data collectors and our study participants along with their parents.

## Author contributions

**Conceptualization:** Bilal Ahmad Rahimi, Aziz Ahmad Khalid, Wahid Ahmad Khalid, Javed Ahmad Rahimi, Walter R Taylor.

**Data curation:** Bilal Ahmad Rahimi, Aziz Ahmad Khalid, Wahid Ahmad Khalid, Javed Ahmad Rahimi.

**Formal analysis:** Bilal Ahmad Rahimi, Aziz Ahmad Khalid.

**Funding acquisition:** Bilal Ahmad Rahimi.

**Investigation:** Bilal Ahmad Rahimi, Aziz Ahmad Khalid, Wahid Ahmad Khalid, Javed Ahmad Rahimi.

**Methodology:** Bilal Ahmad Rahimi.

**Project administration:** Bilal Ahmad Rahimi, Aziz Ahmad Khalid.

**Resources:** Bilal Ahmad Rahimi, Aziz Ahmad Khalid, Wahid Ahmad Khalid, Javed Ahmad Rahimi.

**Software:** Bilal Ahmad Rahimi, Aziz Ahmad Khalid.

**Supervision:** Bilal Ahmad Rahimi, Aziz Ahmad Khalid.

**Validation:** Bilal Ahmad Rahimi.

**Visualization:** Bilal Ahmad Rahimi, Wahid Ahmad Khalid.

**Writing – original draft:** Bilal Ahmad Rahimi.

**Writing – review & editing:** Bilal Ahmad Rahimi, Aziz Ahmad Khalid, Wahid Ahmad Khalid, Javed Ahmad Rahimi, Walter R Taylor.

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
