## [Decision Letter · Decision Letter 0]

13 Jan 2025

PONE-D-24-38181Prevalence and associated factors of overweight and obesity among Afghan school children: a cross-sectional analytical study from Kandahar city, Afghanistan.PLOS ONE

Dear Dr. Rahimi,

Thank you for submitting your manuscript to PLOS ONE. After careful consideration, we feel that it has merit but does not fully meet PLOS ONE’s publication criteria as it currently stands. Therefore, we invite you to submit a revised version of the manuscript that addresses the points raised during the review process.

We look forward to receiving your revised manuscript.

Kind regards,

Marwa Ramadan

Academic Editor

PLOS ONE

**Journal Requirements:**

Prevalence and associated risk factors of stunting, wasting/thinness, and underweight among primary school children in Kandahar City, Afghanistan: a cross-sectional analytical study - https://doi.org/10.1186/s12889-024-19858-z

Prevalence and Associated Factors of Overweight and Obesity among Primary School Children Aged 7–17 Years in Urban Mbarara, Uganda -  https://doi.org/10.1155/2024/5175550

(Among others)

In your revision ensure you cite all your sources (including your own works), and quote or rephrase any duplicated text outside the methods section. Further consideration is dependent on these concerns being addressed.

**Additional Editor Comments:**

This study covers an important topic, and I appreciate the effort involved in conducting research in such a challenging context. However, I have a few points that I would appreciate further clarification on:

-Inclusion of Girls Aged 12–18:

I noticed that the study includes girls aged 12–18. Given the restrictions on girls' education in Afghanistan following the Taliban's takeover, I am wondering how participants in this age group were recruited.

Could you kindly provide more details on how data collection was conducted for this group? I am unsure whether the current context might have posed challenges for including these participants.

-Reliability of Family Income Data:

-I noticed that family income data were collected directly from the child participants. I am uncertain if this approach could affect the accuracy of the data.

Would you be able to explain why this method was selected and whether any steps were taken to confirm the information with parents or guardians?

If verification was not possible, it might be helpful to acknowledge this as a limitation and discuss how it could influence the study’s findings.

Reviewers' comments:

Reviewer's Responses to Questions

**Comments to the Author**

1. Is the manuscript technically sound, and do the data support the conclusions?

Reviewer #1: Partly

Reviewer #2: Yes

2. Has the statistical analysis been performed appropriately and rigorously? 

Reviewer #1: I Don't Know

Reviewer #2: Yes

3. Have the authors made all data underlying the findings in their manuscript fully available?

Reviewer #1: No

Reviewer #2: No

4. Is the manuscript presented in an intelligible fashion and written in standard English?

Reviewer #1: Yes

Reviewer #2: Yes

5. Review Comments to the Author

**Reviewer #1:**  The authors mentioned , being boy, student of private school and belonging to rich families as risk factors associated with overweight and obesity. It is not clear why these are considered as risk factor. Please provide more detail and clarification.

**Reviewer #2: ** This is a good study conducted thoroughly in an adequate sample size; the findings highlight the iceberg of noncommunicable diseases in future generations of Afgasthan. There are no major issues for revision, but some points that can be helpful to make the article more significant are as follows:

In abstract

1. Since the AOR is calculated, it should be multiple logistic regression in statistical analysis.

2. Mean better come with SD

3. In result, numbering started from ii; its better to remove numbering

4. It will be better to have a concise conclusion based on the findings provided in the result.

In body of manuscript:

1. Participants were from the age of 6 years; were they able to provide information in the interview?

2. Total number of schools from which 10 schools were selected? Explain clearly about the calculation of sample size and sampling process step-wise, which will make it easier if readers want to replicate the study.

3. In line 129, "a researcher-made questionnaire was used in two local languages."- its great that the tool was translated into the local languages, but can you explain the process of toll validation?

4. Each table should be described separately in the result section; use 'n' to denote sample instead of N. There's no need to mention numbers when percentages are mentioned because these numbers are mentioned in tables. Information in the description should be mentined in details in the table.

5. * Family which earn <155 Afghanis (<2.15 USD) per person per day is not clear to which information you are referring to.

6. Did you use Chi-square to calculate COR and then logistic regression to calculate AOR?

7. It's better to explain AOR in an understandable way for readers than just mentioning the numbers.

8. Typing error in 237 line and reference no. 27

6. PLOS authors have the option to publish the peer review history of their article (what does this mean? ). If published, this will include your full peer review and any attached files.

**Do you want your identity to be public for this peer review?** For information about this choice, including consent withdrawal, please see our Privacy Policy .

Reviewer #1: No

Reviewer #2: No

---

## [Author Response · Author response to Decision Letter 1]

16 Jan 2025

Response to Reviewers

Journal Requirements:

Please ensure that your manuscript meets PLOS ONE's style requirements, including those for file naming. The PLOSONE style templates can be found at

Answer: Many thanks. Now all the title and body of manuscript is formatted based on the PLOS ONE's style requirements.

Prevalence and associated risk factors of stunting, wasting/thinness, and underweight among primary school children in Kandahar City, Afghanistan: a cross-sectional analytical study - https://doi.org/10.1186/s12889-024-19858-z

Prevalence and Associated Factors of Overweight and Obesity among Primary School Children Aged 7–17 Years in Urban Mbarara, Uganda - https://doi.org/10.1155/2024/5175550

(Among others)In your revision ensure you cite all your sources (including your own works), and quote or rephrase any duplicated text outside the methods section. Further consideration is dependent on these concerns being addressed.

Answer: Many thanks. Now the overlapping text has been corrected accordingly. All the changes have been brought as per comments.

Answer: Many thanks. OK, now they are identical.

Answer: Many thanks. OK, it has been deleted from the other sections.

Answer:

Additional Editor Comments:

This study covers an important topic, and I appreciate the effort involved in conducting research in such a challenging context. However, I have a few points that I would appreciate further clarification on:

-Inclusion of Girls Aged 12–18:

I noticed that the study includes girls aged 12–18. Given the restrictions on girls' education in Afghanistan following the Taliban's takeover, I am wondering how participants in this age group were recruited.

Could you kindly provide more details on how data collection was conducted for this group? I am unsure whether the current context might have posed challenges for including these participants.

Answer: Many thanks. It is an important point. For this, I want to clarify two things: (i) Girls and boys have separate schools in Afghanistan, even before the advent of Taliban. (ii) Taliban have banned girls’ education from grades 7–12, and university. Girls can go to schools from grades 1–6. We have many girls aged 12–18 years going to girls’ schools. Most of these girls are in grade 6 and many girls have repeating the same grade 6 two or three times. This is because the girls and their parents prefer that instead of staying at home, girls should go to school and repeat the same 6th grade two or more times.

-Reliability of Family Income Data:

-I noticed that family income data were collected directly from the child participants. I am uncertain if this approach could affect the accuracy of the data.

Would you be able to explain why this method was selected and whether any steps were taken to confirm the information with parents or guardians?

If verification was not possible, it might be helpful to acknowledge this as a limitation and discuss how it could influence the study’s findings.

Answer: Many thanks. Yes, it affects the reliability. Sorry for not clarifying this properly in the manuscript. In fact, we did not take interviews only from children. We also involved their parents/guardians in interviews, especially if the children were young enough to answer the questions and also about the variables, such as family economic status, information about parents (obesity, hypertension, diabetes mellitus, etc.), and eating habits of the children. Now this information is made clear and added in the manuscript.

Reviewers' comments:

Reviewer's Responses to Questions

Comments to the Author

1. Is the manuscript technically sound, and do the data support the conclusions?

The manuscript must describe a technically sound piece of scientific research with data that supports the conclusions.

Experiments must have been conducted rigorously, with appropriate controls, replication, and sample sizes. The conclusions must be drawn appropriately based on the data presented.

Reviewer #1: Partly

Reviewer #2: Yes

2. Has the statistical analysis been performed appropriately and rigorously?

Reviewer #1: I Don't Know

Reviewer #2: Yes

3. Have the authors made all data underlying the findings in their manuscript fully available?

Reviewer #1: No

Reviewer #2: No4. Is the manuscript presented in an intelligible fashion and written in standard English?

Reviewer #1: Yes

Reviewer #2: Yes

5. Review Comments to the Author

Reviewer #1: The authors mentioned, being boy, student of private school and belonging to rich families as risk factors associated with overweight and obesity. It is not clear why these are considered as risk factor. Please provide more detail and clarification.

Answer: Many thanks for the nice comment. Now I provided more details and clarification for this in the materials and methods section as follows: “A Chi-square test (using crude odds ratio [COR]) was performed to find the association between overweight/obesity and independent variables (suspected risk factors). All the independent variables that were statistically significant in Chi-square analysis were adjusted and assessed for independence in multiple logistic regression (using adjusted odds ratio [AOR]) to determine the risk factors of overweight/obesity among school children”.

Reviewer #2: This is a good study conducted thoroughly in an adequate sample size; the findings highlight the iceberg of noncommunicable diseases in future generations of Afghanistan. There are no major issues for revision, but some points that can be helpful to make the article more significant are as follows:

In abstract

1. Since the AOR is calculated, it should be multiple logistic regression in statistical analysis.

Answer: Many thanks. OK, now it has been corrected as “multiple logistic regression” in statistical analysis.

2. Mean better come with SD

Answer: Many thanks. OK, not SD has been added.

3. In result, numbering started from ii; its better to remove numbering

Answer: Many thanks. Now, the numbering has been removed.

4. It will be better to have a concise conclusion based on the findings provided in the result.

Answer: Many thanks. OK, now the conclusion is made concise based on the findings of our research.

In body of manuscript:

1. Participants were from the age of 6 years; were they able to provide information in the interview?

Answer: Thanks for the great point. In such children, data were collected from their parents/guardians. This issue has been made clear by adding the information as follows: “Data of the young age school children who were not able to provide complete information in the interview were collected from their parents/guardians.”.

2. Total number of schools from which 10 schools were selected? Explain clearly about the calculation of sample size and sampling process step-wise, which will make it easier if readers want to replicate the study.

Answer: Many thanks. There are a total of 145 schools in Kandahar City. Now the suggested information has been added in the materials and methods section as follows: Among the 145 schools in Kandahar City, 10 schools (five government and five private schools) were randomly selected for this study using the lottery method of randomization.

The sample size was calculated using the software of Epi Info version 7.2.2.6 (CDC, Atlanta, Georgia, USA). The two-sided confidence level was taken as 95%, while the power of the study was 90%. A non-response rate of 20% was added. So, our sample size was 2395 school children, with 240 study participants selected from each school. Among these children, 114 (4.8%) were excluded because their parents/guardians refused to take part in the study. The response rate was 95.2%. So, we collected data from 2281 school children”.

3. In line 129, "a researcher-made questionnaire was used in two local languages."- its great that the tool was translated into the local languages, but can you explain the process of toll validation?

Answer: Many thanks for mentioning the important point. We did not explain the process of translation and validation. Now we have explained the process as follows: Rigorous translations of the questionnaire from English into Pashto and Dari languages were undertaken, Experienced translators, who were native Pashto speakers, were selected to independently translate the DASS-21 items and instructions, ensuring clarity and cultural relevance while documenting translation choices and challenges. These initial translations were then thoroughly reviewed by a panel of four bilingual experts, including psychologists and linguists, to resolve discrepancies and ensure that the intended meanings were preserved. Subsequently, a separate bilingual translator conducted a back-translation of the Pashto version into English to identify and address any mistranslations, with the expert panel meticulously evaluating and correcting these issues. Finally, cognitive interviews with Pashto-speaking participants were conducted to evaluate the comprehensibility, clarity, and cultural appropriateness of the translated items, with revisions made as needed.

4. Each table should be described separately in the result section; use 'n' to denote sample instead of N. There's no need to mention numbers when percentages are mentioned because these numbers are mentioned in tables. Information in the description should be mentioned in details in the table.

Answer: Many thanks. Now each table has been described separately in the results section. Now 'n' is used to denote sample instead of N. Now numbers have been removed and only percentages have been used. Information in the description is mentioned in details in the tables.

5. * Family which earn <155 Afghanis (<2.15 USD) per person per day is not clear to which information you are referring to.

Answer: Many thanks. This was a typing error. Now removed.

6. Did you use Chi-square to calculate COR and then logistic regression to calculate AOR?

Answer: Many thanks. Yes, it is mentioned in the materials and methods section as follows: A Chi-square test (using crude odds ratio [COR]) was performed to find the association between overweight/obesity and independent variables (suspected risk factors). All the independent variables that were statistically significant in Chi-square analysis were adjusted and assessed for independence in multiple logistic regression (using adjusted odds ratio [AOR]) to determine the risk factors of overweight/obesity among school children.

7. It's better to explain AOR in an understandable way for readers than just mentioning the numbers.

Answer: Many thanks. OK, now AOR is mentioned in an understandable way in the materials and methods section.

8. Typing error in 237 line and reference no. 27

Answer: Many thanks. Now the typing errors have been corrected.

---

## [Decision Letter · Decision Letter 1]

13 Feb 2025

Prevalence and associated factors of overweight and obesity among Afghan school children: a cross-sectional analytical study from Kandahar city, Afghanistan.

PONE-D-24-38181R1

Dear Dr. Bilal Ahmad Rahimi

We’re pleased to inform you that your manuscript has been judged scientifically suitable for publication and will be formally accepted for publication once it meets all outstanding technical requirements.

Kind regards,

Marwa Ramadan

Academic Editor

PLOS ONE

Additional Editor Comments (optional):

Reviewers' comments:

Reviewer's Responses to Questions

**Comments to the Author**

1. If the authors have adequately addressed your comments raised in a previous round of review and you feel that this manuscript is now acceptable for publication, you may indicate that here to bypass the “Comments to the Author” section, enter your conflict of interest statement in the “Confidential to Editor” section, and submit your "Accept" recommendation.

Reviewer #1: All comments have been addressed

Reviewer #2: All comments have been addressed

2. Is the manuscript technically sound, and do the data support the conclusions?

Reviewer #1: Yes

Reviewer #2: Yes

3. Has the statistical analysis been performed appropriately and rigorously? 

Reviewer #1: Yes

Reviewer #2: Yes

4. Have the authors made all data underlying the findings in their manuscript fully available?

Reviewer #1: Yes

Reviewer #2: No

5. Is the manuscript presented in an intelligible fashion and written in standard English?

Reviewer #1: Yes

Reviewer #2: Yes

6. Review Comments to the Author

Reviewer #1: The publication brings another aspect of the malnutrition (overweight) in Afghanistan, specifically in Kandahar province. I have no major comments

Reviewer #2: (No Response)

7. PLOS authors have the option to publish the peer review history of their article (what does this mean? ). If published, this will include your full peer review and any attached files.

**Do you want your identity to be public for this peer review?** For information about this choice, including consent withdrawal, please see our Privacy Policy .

Reviewer #1: No

Reviewer #2: **Yes: ** Laxmi Gautam

---

## [Editor Report · Acceptance letter]

PONE-D-24-38181R1

PLOS ONE

Dear Dr. Rahimi,

I'm pleased to inform you that your manuscript has been deemed suitable for publication in PLOS ONE. Congratulations! Your manuscript is now being handed over to our production team.

Kind regards,

on behalf of

Dr. Marwa Ramadan

Academic Editor

PLOS ONE